# How deep convolutional neural networks lose spatial information with training

## Abstract

A central question of machine learning is how deep nets manage to learn tasks in high dimensions. An appealing hypothesis is that they achieve this feat by building a representation of the data where information irrelevant to the task is lost. For image datasets, this view is supported by the observation that after (and not before) training, the neural representation becomes less and less sensitive to diffeomorphisms acting on images as the signal propagates through the net. This loss of sensitivity correlates with performance, and surprisingly correlates with a *gain* of sensitivity to white noise acquired during training. These facts are unexplained, and as we demonstrate still hold when white noise is added to the images of the training set. Here, we *(i)* show empirically for various architectures that stability to image diffeomorphisms is achieved by both spatial and channel pooling, *(ii)* introduce a model scale-detection task which reproduces our empirical observations on spatial pooling and *(iii)* compute analitically how the sensitivity to diffeomorphisms and noise scales with depth due to spatial pooling. The scalings are found to depend on the presence of strides in the net architecture. We find that the increased sensitivity to noise is due to the perturbing noise piling up during pooling, after being rectified by ReLU units.

## 1 Introduction

Deep learning algorithms can be successfully trained to solve a large variety of tasks (Amodei et al., 2016; Huval et al., 2015; Mnih et al., 2013; Shi et al., 2016; Silver et al., 2017), often revolving around classifying data in high-dimensional spaces. If there was little structure in the data, the learning procedure would be cursed by the dimension of these spaces: achieving good performances would require an astronomical number of training data (Luxburg & Bousquet, 2004). Consequently, real datasets must have a specific internal structure that can be learned with fewer examples. It has been then hypothesized that the effectiveness of deep learning lies in its ability of building 'good' representations of this internal structure, which are insensitive to aspects of the data not related to the task (Ansuini et al., 2019; Shwartz-Ziv & Tishby, 2017; Recanatesi et al., 2019), thus effectively reducing the dimensionality of the problem.

In the context of image classification, Bruna & Mallat (2013); Mallat (2016) proposed that neural networks lose irrelevant information by learning representations that are insensitive to small deformations of the input, also called diffeomorphisms. This idea was tested in modern deep networks by Petrini et al. (2021), who introduced the following measures

$$D_f = \frac{\mathbb{E}_{x,\tau}\|f(\tau(x)) - f(x)\|^2}{\mathbb{E}_{x_1,x_2}\|f(x_1) - f(x_2)\|^2}, \qquad G_f = \frac{\mathbb{E}_{x,\eta}\|f(x + \eta) - f(x)\|^2}{\mathbb{E}_{x_1,x_2}\|f(x_1) - f(x_2)\|^2}, \qquad R_f = \frac{D_f}{G_f}, \qquad (1)$$

to probe the sensitivity of a function $f$—either the output or an internal representation of a trained network—to random diffeomorphisms $\tau$ of $x$ (see example in Fig. 1, left), to large white noise perturbations $\eta$ of magnitude $\|\tau(x) - x\|$, and in relative terms, respectively. Here the input images $x$, $x_1$ and $x_2$ are sampled uniformly from the test set. In particular, the test error of trained networks is correlated with $D_f$ when $f$ is the network output. Less intuitively, the test error is anti-correlated with the sensitivity to white noise $G_f$. Overall, it is the relative sensitivity $R_f$ which correlates best with the error (Fig. 1, middle). This correlation is learned over training—as it is not seen at initialization—and built up layer by layer (Petrini et al., 2021). These phenomena are not simply due

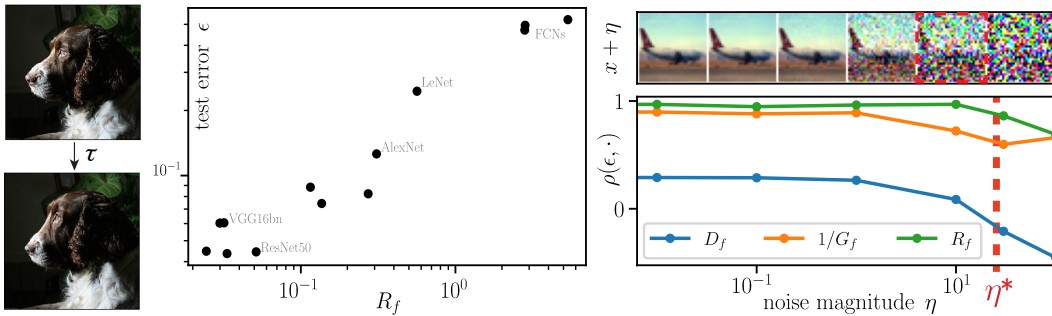

Figure 1: Left: example of a random diffeomorphism $\tau$ applied to an image. Center: test error vs relative sensitivity to diffeomorphisms of the predictor for a set of networks trained on CIFAR10, adapted from Petrini et al. (2021). Right: Correlation coefficient between test error $\epsilon$ and $D_f$, $G_f$ and $R_f$ when training different architectures on noisy CIFAR10, $\rho(\epsilon, X) = \text{Cov}(\log \epsilon, \log X)/\sqrt{\text{Var}(\log \epsilon)\text{Var}(\log X)}$. Increasing noise magnitudes are shown on the $x$-axis and $\eta^* = \mathbb{E}_{\tau,x}\|\tau(x) - x\|^2$ is the one used for the computation of $G_f$. Samples of a noisy CIFAR10 datum are shown on top. Notice that $D_f$ and particularly $R_f$ are positively correlated with $\epsilon$, whilst $G_f$ is negatively correlated with $\epsilon$. The corresponding scatter plots are in Fig. 10 (appendix).

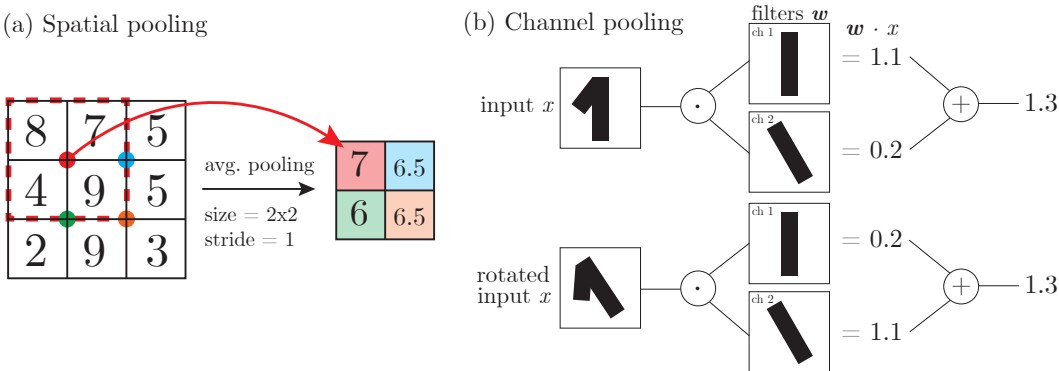

Figure 2: Spatial vs. channel pooling. (a) Spatial average pooling (size 2x2, stride 1) computed on a representation of size 3x3. One can notice that nearby pixel variations are smaller after pooling. (b) If the filters of different channels are identical up to e.g. a rotation of angle $\theta$, then, averaging the output of the application of such filters makes the result invariant to input rotations of $\theta$. This averaging is an example of channel pooling.

to benchmark data being noiseless, as they persist when input images are corrupted by some small noise (Fig. 1, right).

Operations that grant insensitivity to diffeomorphisms in a deep network have been identified previously (e.g. Goodfellow et al. (2016), section 9.3, sketched in Fig. 2). The first, *spatial* pooling, integrates local patches within the image, thus losing the exact location of its features. The second, *channel* pooling, requires the interaction of different channels, which allows the network to become invariant to any local transformation by properly learning filters that are transformed versions of one another. However, it is not clear whether these operations are actually learned by deep networks and how they conspire in building good representations. Here we tackle this question by unveiling empirically the emergence of spatial and channel pooling, and disentangling their role. Below is a detailed list of our contributions.

## 1.1 OUR CONTRIBUTIONS

- We disentangle the role of spatial and channel pooling within deep networks trained on CIFAR10 (Section 2). More specifically, our experiments reveal the significant contribution of spatial pooling in decreasing the sensitivity to diffeomorphisms.

- In order to isolate the contribution of spatial pooling and quantify its relation with the sensitivities to diffeomorphism and noise, we introduce idealized scale-detection tasks (Section 3). In these tasks, data are made of two active pixels and classified according to their distance. We find the same correlations between test error and sensitivities of trained networks as found in Petrini et al. (2021). In addition, the neural networks which perform the best on real data tend to be the best on these tasks.

- We theoretically analyze how simple CNNs, made by stacking convolutional layers with filter size $F$ and stride $s$, learn these tasks (Section 4). We find that the trained networks perform spatial pooling for most of its layers. We show and verify empirically that the sensitivities $D_k$ and $G_k$ of the $k$-th hidden layer follow $G_k \sim A_k$ and $D_k \sim A_k^{-\alpha_s}$, where $A_k$ is the effective receptive field size and $\alpha_s = 2$ if there is no stride, $\alpha_s = 1$ otherwise.

The code and details for reproducing experiments are available online at tinyurl.com/github-experiments.

## 1.2 RELATED WORK

In the neuroscience literature, the understanding of the relevance of pooling in building invariant representations dates back to the pioneering work of Hubel & Wiesel (1962). By studying the cat visual cortex, they identified two different kinds of neurons: simple cells responding to e.g. edges at specific angles and complex cells that *pool* the response of simple cells and detect edges regardless of their position or orientation in the receptive field. More recent accounts of the importance of learning invariant representations in the visual cortex can be found in Niyogi et al. (1998); Anselmi et al. (2016); Poggio & Anselmi (2016).

In the context of artificial neural networks, layers jointly performing spatial pooling and strides have been introduced with the early CNNs of Lecun et al. (1998), following the intuition that local averaging and subsampling would reduce the sensitivity to small input shifts. Ruderman et al. (2018) investigated the role of spatial pooling and showed empirically that networks with and without pooling layers converge to similar deformation stability, suggesting that spatial pooling can be learned in deep networks. In our work, we further expand in this direction by jointly studying diffeomorphisms and noise stability and proposing a theory of spatial pooling for a simple task.

The depth-wise loss of irrelevant information in deep networks has been investigated by means of the information bottleneck framework (Shwartz-Ziv & Tishby, 2017; Saxe et al., 2019) and the intrinsic dimension of the networks internal representations (Ansuini et al., 2019; Recanatesi et al., 2019). However, these works do not specify what is the irrelevant information to be disregarded, nor the mechanisms involved in such a process.

The stability of trained networks to noise is extensively studied in the context of adversarial robustness (Fawzi & Frossard, 2015; Kanbak et al., 2018; Alcorn et al., 2019; Alaifari et al., 2018; Athalye et al., 2018; Xiao et al., 2018a; Engstrom et al., 2019). Notice that our work differs from this literature by the fact that we consider typical perturbations instead of worst-case ones.

## 2 EMPIRICAL OBSERVATIONS ON REAL DATA

In this section we analyze the parameters of deep CNNs trained on CIFAR10 and ImageNet, so as to understand how they build representations insensitive to diffeomorphisms (details of the experiments in App. B). The analysis builds on two premises, the first being the assumption that insensitivity is built layer by layer in the network, as shown in Fig. 3. Hence, we focus on how each of the layers in a deep network contribute towards creating an insensitive representation. More specifically, let us denote with $f_k(x)$ the internal representation of an input $x$ at the $k$-th layer of the network. The entries of $f_k$ have three indices, one for the channel $c$ and two for the spatial location $(i, j)$. The relation between $f_k$ and $f_{k-1}$ is the following,

$$[f_k(x)]_{c;i,j} = \phi \left( b_c^k + \sum_{c'=1}^{H_{k-1}} \boldsymbol{w}_{c,c'}^k \cdot \boldsymbol{p}_{i,j} \left( [f_{k-1}(x)]_{c'} \right) \right) \quad \forall c = 1, \ldots, H_k, \qquad (2)$$

where: $H_k$ denotes the number of channels at the $k$-th layer; $b_c^k$ and $\boldsymbol{w}_{c,c'}^k$ the biases and *filters* of the $k$-th layer; each filter $\boldsymbol{w}_{c,c'}^k$ is a $F \times F$ matrix with $F$ the filter size; $\boldsymbol{p}_{i,j}\left([f_{k-1}(x)]_{c'}\right)$ denotes a $F \times F$-dimensional patch of $[f_{k-1}(x)]_{c'}$ centered at $(i,j)$; $\phi$ the activation function. The second premise is that a general diffeomorphism can be represented as a displacement field over the image, which indicates how each pixel moves in the transformation. Locally, this displacement field can be decomposed into a constant term and a linear part: the former corresponds to local translations, the latter to stretchings, rotations and shears.[1]

**Invariance to translations via spatial pooling.** Due to weight sharing, i.e. the fact that the same filter $\boldsymbol{w}_{c,c'}^k$ is applied to all the local patches $(i,j)$ of the representation, the output of a convolutional layer is *equivariant* to translations by construction: a shift of the input is equivalent to a shift of the output. To achieve an *invariant* representation it suffices to sum up the spatial entries of $f_k$—an operation called pooling in CNNs, we refer to it as *spatial* pooling to stress that the sum runs over the spatial indices of the representation. Even if there are no pooling layers at initialization, they can be realized by having homogeneous filters, i.e. all the $F \times F$ entries of $\boldsymbol{w}_{c,c'}^{k+1}$ are the same. Therefore, the closer the filters are to the homogeneous filter, the more they decrease the sensitivity of the representation to local translations.

**Invariance to other transformations via channel pooling.** The example of translations shows that building invariance can be performed by constructing an equivariant representation, and then pooling it. Invariance can also be built by pooling *across* channels. A two-channel example is shown Fig. 2, panel (b), where the filter of the second channel is built so as to produce the same output as the first channel when applied to a rotated input. The same idea can be applied more generally, e.g. to the other components of diffeomorphisms—such as local stretchings and shears. Below, we refer generically to any operation that build invariance to diffeomorphisms by assembling distinct channels as *channel pooling*.

**Disentangling spatial and channel pooling.** The relative sensitivity to diffeomorphisms $R_k$ of the $k$-th layer representation $f_k$ decreases after each layer, as shown in Fig. 3. This implies that spatial or channel pooling are carried out along the whole network. To disentangle their contribution we perform the following experiment: shuffle at random the connections between channels of successive convolutional layers, while keeping the weights unaltered. Channel shuffling amounts to randomly permuting the values of $c, c'$ in Eq. 2, therefore it breaks any channel pooling while not affecting single filters. The values of $R_k$ for deep networks after channel shuffling are reported in Fig. 3 as dashed lines and compared with the original values of $R_k$ in full lines. If only spatial pooling was present in the network, then the two curves would overlap. Conversely, if the decrease in $R_k$ was all due to the interactions between channels, then the shuffled curves should be constant. Given that neither of these scenarios arises, we conclude that both kinds of pooling are being performed.

**Emergence of spatial pooling after training.** To bolster the evidence for the presence of spatial pooling, we analyze the filters of trained networks. Since spatial pooling can be built by having homogeneous filters, we test for its presence by looking at the frequency content of learned filters $\boldsymbol{w}_{i,j}^k$. In particular, we consider the average squared projection of filters onto "Fourier modes" $\{\Psi_l\}_{l=1,...,F^2}$, taken as the eigenvectors of the discrete Laplace operator on the $F \times F$ filter grid. The square projections averaged over channels read

$$\gamma_{k,l} = \frac{1}{H_{k-1}H_k} \sum_{c=1}^{H_k} \sum_{c'=1}^{H_{k-1}} \left[\Psi_l \cdot \boldsymbol{w}_{c,c'}^k\right]^2, \tag{3}$$

and are shown in Fig. 4, 1$^{\text{st}}$ and 2$^{\text{nd}}$ row. When training a deep network such as VGG11 (with and without batch-norm) (Simonyan & Zisserman, 2015) on CIFAR10, filters of layers 2 to 6 become low-frequency with training, while layers 1, 7, 8 do not. Accordingly, larger gaps between dashed and full lines in Fig. 3 (right) open at layer 1, 7, 8: reduction in sensitivity is not due to spatial pooling in these layers. Moreover, the fact that the two dashed curves overlap is consistent with the frequency

---

[1] The displacement field around a pixel $(u_0, v_0)$ is approximated as $\tau(u,v) \simeq \tau(u_0, v_0) + J(u_0, v_0)[u - u_0, v - v_0]^T$, where $\tau(u_0, v_0)$ corresponds to translations and $J$ is the Jacobian matrix of $\tau$ whose trace, antisymmetric and symmetric traceless parts correspond to stretchings, rotations and shears, respectively.

content of filters being the same for the two architectures after training. In the case of ImageNet, filters at all layers become low-frequency, except for $k = 1$.

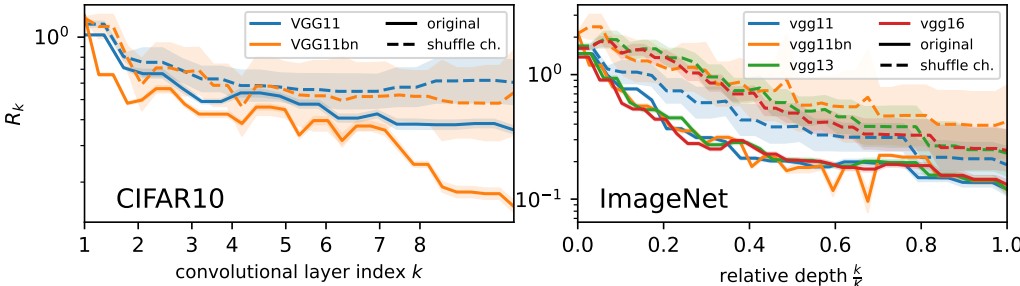

Figure 3: Relative sensitivity $R_k$ as a function of depth for VGG architectures trained on CIFAR10 (left) and ImageNet (right). Full lines refer to the original networks, dashed lines to the ones with shuffled channels. $K$ is the total depth of the networks. Experiments with different architectures are reported in App. C.

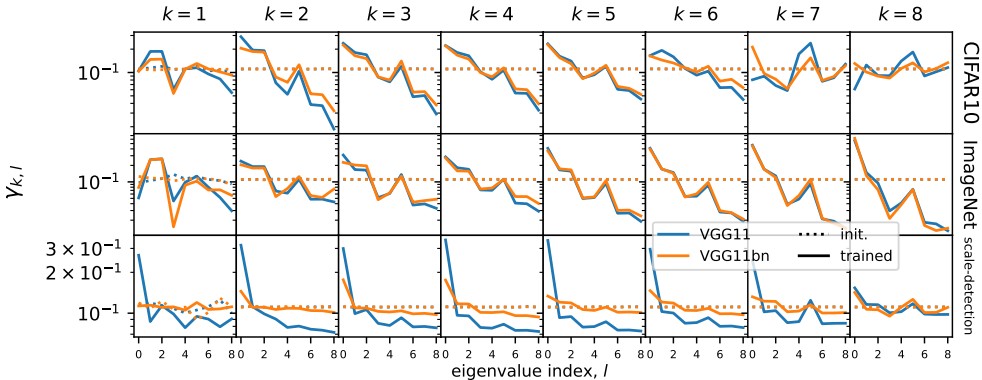

Figure 4: Projections of the network filters for VGG11 and VGG11bn onto the 9 eigenvectors of the $(3 \times 3)$-grid Laplacian when training on CIFAR10 (1st row), ImageNet, (2nd row) and the scale-detection task (3rd row): dotted and full lines correspond to initialization and trained networks, respectively. The $x$-axis reports low to high frequencies from left to right. Deeper layers are reported in rightmost panels. Low-frequency modes are the dominant components in layers 2-6 when training on CIFAR10, in layers 2-8 for ImageNet. The first (constant) mode has most of the power throughout the network for scale-detection task 1. An aggregate measure of the spatial frequency content of filters is reported in App. C, Fig. 12.

## 3 SIMPLE SCALE-DETECTION TASKS CAPTURE REAL-DATA OBSERVATIONS

To sum up, the empirical evidence presented in Section 2 indicates that *(i)* the generalization performance of deep CNNs correlates with their insensitivity to diffeomorphisms and sensitivity to Gaussian noise (Fig. 1); *(ii)* deep CNNs build their sensitivities layer by layer via spatial and channel pooling. We introduce now two idealized scale-detection tasks where the phenomena *(i)* and *(ii)* emerge again, and we can isolate the contribution of spatial pooling. Given the simpler structure of these tasks with respect to real data, we can understand quantitatively how spatial pooling builds up insensitivity to diffeomorphisms and sensitivity to Gaussian noise, as we show in Section 4.

**Definition of scale-detection tasks.** Consider input images $x$ consisting of two active pixels on an empty background.

**Task 1:** Inputs are classified by comparing the euclidean distance $d$ between the two active pixels and some *characteristic scale* $\xi$, as in Fig. 5, left. Namely, the label is $y = \text{sign}\,(\xi - d)$.

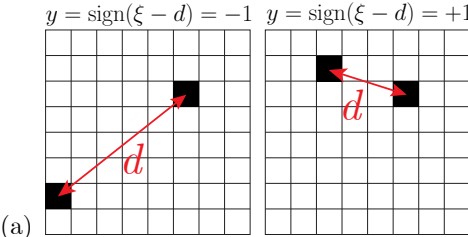 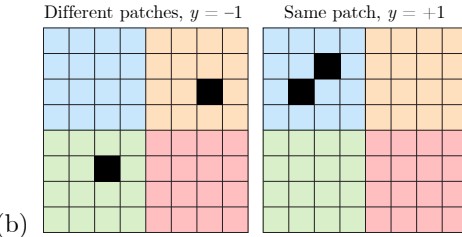

Figure 5: Example inputs for the scale-detection tasks. Task 1 (a): the label depends on whether the euclidean distance $d$ is larger (left) or smaller (right) than the characteristic scale $\xi$. Task 2 (b): the label depends on whether the active pixels belong to the same patch of size $\xi$ (right) or not (left)—patches are shown in different colors.

Notice that a small diffeomorphism of such images corresponds to a small displacement of the active pixels. Specifically, each of the active pixels is moved to either of its neighboring pixels or left in its original position with equal probability.[2] By introducing a gap $g$ such that $d \in [\xi - g/2, \xi + g/2]$, task 1 becomes invariant to displacements of size smaller than $g$. Therefore, we expect that a neural network trained on task 1 will lose any information on the exact location of the active pixels within the image, thus becoming insensitive to diffeomorphisms. Intuitively, spatial pooling up to the scale $\xi$ is the most direct mean to achieve such insensitivity. The result of the integration depends on whether none, one or both the active pixels lie within the pooling window, thus it is still informative of the task. We will show empirically that this is indeed the solution reached by trained CNNs.

**Task 2:** Inputs are partitioned into nonoverlapping patches of size $\xi$, as in Fig. 5, right. The label $y$ is $+1$ if the active pixels fall within the same patch, $-1$ otherwise.

In task 2, the irrelevant information is the location of the pixels within each of the non-overlapping patches. The simplest means to lose such information requires to couple spatial pooling with a stride of the size of the pooling window itself.

**Same phenomenology as in real image datasets.** Although these scale-detection tasks are much simpler than standard benchmark datasets, deep networks trained on task 1 display the same phenomenology highlighted in Section 2 for networks trained on CIFAR10 and ImageNet. First, the test error is positively correlated with the sensitivity to diffeomorphisms of the network predictor (Fig. 8, left panel, in App. C) and negatively correlated with its sensitivity to Gaussian noise (middle panel) for a whole range of architectures. As a result, the error correlates well with the relative sensitivity $R_f$ (right panel). Secondly, the internal representations of trained networks $f_k$ become progressively insensitive to diffeomorphisms and sensitive to Gaussian noise through the layers, as shown in Fig. 9 of App. C. Importantly, the curves relating sensitivities to the relative depth remain essentially unaltered if the channels of the networks are shuffled (shown as dashed lines in Fig. 9). We conclude that, on the one hand channel pooling is negligible, and, on the other hand, all channels are approximately equal to the mean channel. Finally, direct inspection of the filters (Fig. 4, bottom row) shows that the 0-frequency component grows much larger than the others over training for layers 1-7, which are the layers where $R_k$ decreases the most in Fig. 9. Filters are thus becoming nearly homogeneous, which means that the convolutional layers become effectively pooling layers.

## 4 THEORETICAL ANALYSIS OF SENSITIVITIES IN SCALE-DETECTION TASKS

We now provide a scaling analysis of the sensitivities to diffeomorphisms and noise in the internal representations of simple CNNs trained on the scale-detection tasks of Section 3. It allows to quantitatively understand how spatial pooling makes the internal representations of the network progressively more insensitive to diffeomorphisms and sensitive to Gaussian noise.

---

[2]We fix the length of these displacements to 1 pixel because *(i)* is the smallest value that prevents the use of pixel interpolation, which would make one active pixel an extended object *(ii)* allows for the analysis of Section 4.

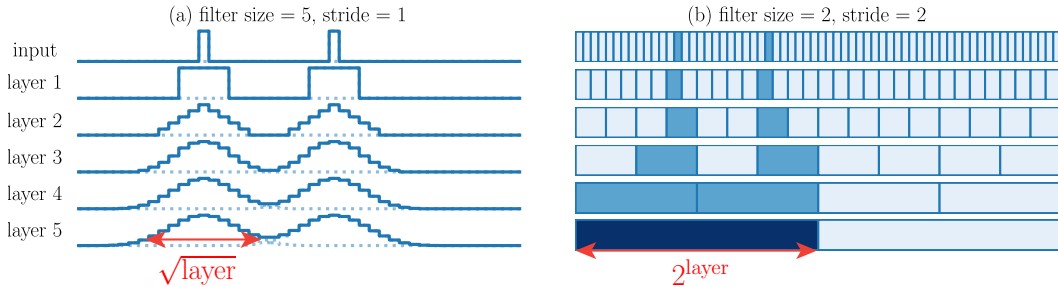

Figure 6: Hidden layers representations of simple CNNs for a scale-detection input for stride $s = 1$ and filter size $F = 5$ (left) and $s = F = 2$ (right) when having homogeneous filters at every layer. The effective receptive field size of the last layer in the two different cases is shown in red. (Left) every active pixel in the input becomes a Gaussian profile whose width increases throughout the network. (Right) every neuron in layer $k$ has activity equal to the number of active pixels which are present in its receptive field of width $2^k$. The dark blue in the last layer indicates that there are 2 active pixels in its receptive field, while the lighter blue of the precedent layers indicates that there is just 1.

**Setup.** We consider simple CNNs made by stacking $\tilde{K}$ identical convolutional layers with generic filter size $F$, stride $s = 1$ or $F$ and ReLU activation function $\phi(x) = \max(0, x)$. In particular, we train CNNs with stride 1 on task 1 and CNNs with stride $F$ on task 2. For the sake of simplicity, we consider the one-dimensional version of the scale-detection tasks, but our analysis carries unaltered to the two-dimensional case. Thus, input images are sequences $x = (x_i)_{i=1,\dots,L}$ of $L$ pixels, where $x_i = 0$ for all pixels except two. For the active pixels $x_i = \sqrt{L/2}$, so that all input images have $\|x\|^2 = L$. We will also consider single-pixel data $\delta_j = (\delta_{j,i})_{i=1,\dots,L}$. If the active pixels in $x$ are the $i$-th and the $j$-th, then $x = \sqrt{L/2}\,(\delta_i + \delta_j)$. For each layer $k$, the internal representation $f_k(x)$ of the trained network is defined as in Eq. 2. The *receptive field* of the $k$-th layer is the number of input pixels contributing to each component of $f_k(x)$. We define the *effective* receptive field $A_k$ as the typical size of the representation of a single-pixel input, $f_k(\delta_i)$, as illustrated in red in Fig. 6. We denote the sensitivities of the $k$-th layer representation with a subscript $k$ ($D_k$ for diffeomorphisms, $G_k$ for noise, $R_k$ for relative).

**Assumptions.** All our results are based on the assumption that the first few layers of the trained network behave effectively as a single channel with a homogeneous positive filter and no bias. The equivalence of all the channels with their mean is supported by Fig. 9, which shows how shuffling channels does not affect the internal representations of VGGs. In addition, Fig. 4 (bottom row) shows that the mean filters of the first few layers are nearly homogeneous. We set the homogeneous value of each filter so as to keep the norm of representations constant over layers. Moreover, we implement a deformation of the input $x$ of our scale-detection tasks as a random displacement of each active pixel at either left or wight with probability 1/2.

## 4.1 TASK 1, STRIDE 1

For a CNN with stride 1, under the homogeneous filter assumption, the size of the effective receptive field $A_k$ grows as $\sqrt{k}$. A detailed proof is presented in App. A and Fig. 6, left panel, shows an illustration of the process. Intuitively, applying a homogeneous filter to a representation is equivalent to making each pixel diffuse, i.e. distributing its intensity uniformly over a neighborhood of size $F$. With a single-pixel input $\delta_i$, the effective receptive field of the $k$-th layer $f_k(\delta_i)$ is equivalent to a $k$-step diffusion of the pixel, thus it approaches a Gaussian distribution of standard deviation $\sqrt{k}$ centered at $i$. The size $A_k$ is the standard deviation, thus $A_k \sim \sqrt{k}$. The proof we present in App. A requires large depth $\tilde{K} \gg 1$ and large image width $L \gg F\tilde{K}^{1/2}$ and the empirical studies of Section 3 satisfy these contraints ($F \sim 3$, $L \sim 32$ and $\tilde{K} \sim 10$).

We remark that at initialization, $f_k(x)$ behave, in the limit of large number of channels and width (and small bias), as Gaussian random fields with correlation matrix $\mathbb{E}\left[f_k(x)f_k(y)\right] \approx \delta(x - y)$, with $\delta$ the Dirac delta (Schoenholz et al., 2017; Xiao et al., 2018b). This spiky correlation matrix implies that for any perturbation $y = x + \varepsilon$, the representation $f_k(y)$ changes with respect to $f_k(x)$ independently

on $\varepsilon$. This behavior is remarkably different to the smooth case achieved by the diffusion, after training. Consequently, both $D_k$ and $G_k$ are constant with respect to $k$ at initialization . This is consistent with the observations reported in Fig. 7.

**Sensitivity to diffeomorphisms.** Let $i$ and $j$ denote the active pixels locations, so that $x \propto \delta_i + \delta_j$. Since both the elements of the inputs and those of the filters are non-negative, the presence of ReLU nonlinearities is irrelevant and the first few hidden layers are effectively linear layers. Hence the representations are linear in the input, so that $f_k(x) = f_k(\delta_i + \delta_j) = f_k(\delta_i) + f_k(\delta_j)$. In addition, since the effect of a diffeomorphism is just a 1-pixel translation of the representation irrespective of the original positions of the pixels, the normalized sensitivity $D_k$ can be approximated as follows

$$D_k \sim \frac{\|f_k(\delta_{i+1}) - f_k(\delta_i)\|_2^2}{\|f_k(\delta_i)\|_2^2}. \tag{4}$$

The denominator in Eq. 4 is the squared norm of a Gaussian distribution of width $\sqrt{k}$, $\|f_k(v_i)\|_2^2 \sim k^{-1/2}$. The numerator compares $f_k$ with a small translation of itself, thus it can be approximated by the squared norm of the derivative of the Gaussian distribution, $\|f_k(\delta_{i+1}) - f_k(\delta_i)\|_2^2 \sim k^{-3/2}$. Consequently, we have

$$D_k \sim k^{-1} \sim A_k^{-2}. \tag{5}$$

**Sensitivity to Gaussian noise.** To analyze $G_k$ one must take into account the rectifying action of ReLU, which sets all the negative elements of its input to zero. The first ReLU is applied after the first homogeneous filters, thus the zero-mean noise is superimposed on a patch of $F$ active pixels. Outside such a patch, only positive noise terms survive. Within the patch, being summed to a positive background, also negative terms can survive the rectification of ReLU. Nevertheless, if the size of the image is much larger than the filter size, the contribution from active pixels to $G_k$ is negligible and we can approximate the difference between noisy and original representations $f_1(x + \eta) - f_1(x)$ with the rectified noise $\phi(\eta)$. After the first layer, the representations consist of non-negative numbers, thus we can forget again the ReLU and write

$$G_k \sim \frac{\mathbb{E}_\eta \|f_k(\phi(\eta))\|_2^2}{\|f_k(\delta_i)\|_2^2}. \tag{6}$$

Repeated applications of homogeneous filters to the rectified noise $\phi(\eta)$ result again in a diffusion of the signal. Since $\phi(\eta)$ has different independent and identically distributed non-zero entries for different realizations of $\eta$, averaging over $\eta$ is equivalent to considering a homogeneous profile for $f_k(\phi(\eta))$. As a result, the numerator in Eq. 6 is a constant independent of $k$. The denominator is the same as in Eq. 4, $\|f_k(\delta_i)\|_2^2 \sim k^{-1/2}$, hence

$$G_k \sim k^{1/2} \sim A_k, \tag{7}$$

i.e. the sensitivity to Gaussian noise grows as the size of the effective receptive fields. From the ratio of Eq. 5 and Eq. 7, we get $R_k \sim A_k^{-3}$.

## 4.2 TASK 2, STRIDE EQUAL FILTER SIZE

When the stride $s$ equals to the filter size $F$ the number of pixels of the internal representations is reduced by a factor $F$ at each layer, thus $f_k$ consists of $L/F^k$ pixels. Meanwhile, the effective size of the receptive fields grows exponentially at the same rate: $A_k = F^k$ (see Fig. 6, left for an illustration).

**Sensitivity to diffeomorphisms.** For a given layer $k$, consider a partition of the input image into $L/F^k$ patches. Each pixel of $f_k$ only looks at one such patch and its intensity coincides with the number of active pixels within the patch. As a result, the only diffeomorphisms that change $f_k$ are those which move one of the active pixels from one patch to another. Since active pixels move by 1, this can only occur if one of the active pixels was originally located at the border of a patch, which in turn occurs with probability $\sim 1/F^k$. In addition, the norm $\|f_k(\delta_i)\|_2$ at the denominator does not scale with $k$, so that

$$D_k \sim F^{-k} \sim A_k^{-1}. \tag{8}$$

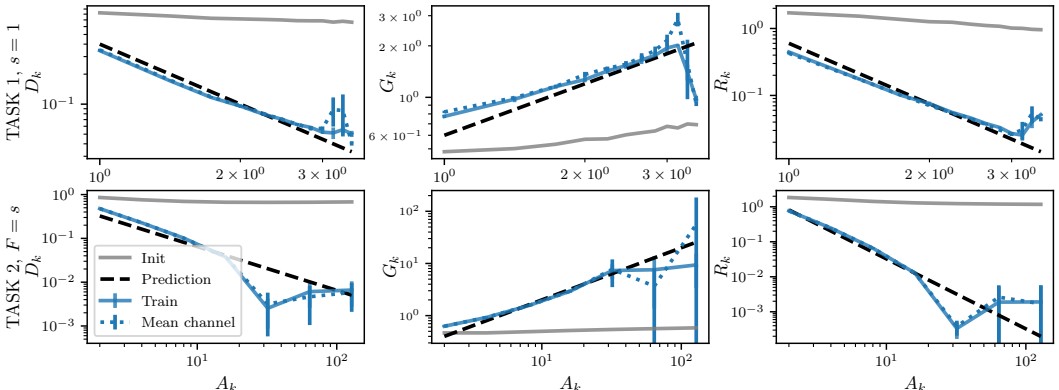

Figure 7: Sensitivities of internal representations $f_k$ of simple CNNs against the $k$-th layer receptive field size $A_k$ for trained networks (solid blue) and at initialization (solid gray). The top row refers to task 1 with $s = 1$ and $F = 3$; the bottom row to task 2 with $F = s = 2$. For a first large part of the network, the sensitivities obtained by replacing each layer with the mean channel (blue dotted) overlap with the original sensitivities. Predictions Eq. 5, Eq. 7 for task 1 and Eq. 8, Eq. 9 for task 2 are shown as black dashed lines.

**Sensitivity to Gaussian noise.** Each pixel of $f_k$ looks at a patch of the input of size $F^k$, thus $f_k$ is affected by the sum of all the noises acting on such patch. Since these noises have been rectified by ReLU, by the Central Limit Theorem the sum scales as the number of summands $F_k$. Thus, the contribution of each pixel of $f_k$ to the numerator of $G_k$ scales as $(F^k)^2$. As there are $L/F^k$ pixels in $f_k$, one has

$$G_k \sim (F^k)^2 \left(L/F^k\right) \sim F^k \sim A_k. \tag{9}$$

Without rectification, the sum of $F^k$ independent noises would scale as the square root of the number of summands $F^k$, yielding a constant $G_k$. We conclude that the rectifying action of ReLU is crucial in building up sensitivity to noise. $R_k \sim A_k^{-2}$ follows from the ratio of Eq. 8 and Eq. 9.

### 4.3 COMPARING PREDICTIONS WITH EXPERIMENTS

We test our scaling predictions (Eq. 5 to Eq. 9) in Fig. 7, for stride 1 CNNs trained on task 1 and stride $F$ CNNs trained on task 2 in the top and bottom panels, respectively. Notice that if all the filters at a given layer are replaced with their average, the behavior of the sensitivities as a function of depth does not change (compare solid and dotted blue curves in the figure). This confirms our assumption that all channels behave like the mean channel. In addition, Tables 1 and 2 show that the mean filters are approximately homogeneous. Further details on the experiments are provided in App. B.

## 5 CONCLUSION

The meaning of an image often depends on sparse regions of the data, as evidenced by the fact that artists only need a small number of strokes to represent a visual scene. The exact locations of the features determining the image class are flexible, and indeed diffeomorphisms of limited magnitude leave the class unchanged. Here, we have shown that such an invariance is learned in deep networks by performing spatial pooling and channel pooling. Modern architectures learn these pooling operations—as they are not imposed by the architecture—suggesting that it is best to let the pooling adapt to the specific task considered. Interestingly, spatial pooling comes together with an increased sensitivity to random noise in the image, as captured in simple artificial models of data.

It is commonly believed that the best architectures are those that extract the features of the data most relevant for the task. The pooling operations studied here, which allow the network to forget the exact locations of these features, are probably more effective when features are better extracted. This point may be responsible for the observed strong correlations between the network performance and its stability to diffeomorphisms. Designing synthetic models of data whose features are combinatorial and stable to smooth transformations is very much needed to clarify this relationship, and ultimately understand how deep networks learn high-dimensional tasks with limited data.

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

APPENDIX

## A  TASK 1, STRIDE 1: PROOFS

In Section 4.1 we consider a simple CNN with stride $s = 1$ and filter size $F$ trained on scale-detection task 1. We fix the total depth of these networks to be $\tilde{K}$. We postulated in Sec. 4 that this network displays a one-channel solution with homogeneous filter $[1/F, ..., 1/F]$ and no bias. We can understand the representation $f_k(x)$ at layer $k$ of an input datum $x$ by using single-pixel inputs $\delta_i$. Let us recall that these inputs have all components to 0 except the $i$-th, set to 1. Then, we have that a general datum $x$ is given by $x \propto (\delta_i + \delta_j)$, where $i$ and $j$ are the locations of the active pixel in $x$. We have argued in the main text that the representation $f_k(\delta_i)$ is a Gaussian distribution with width $\sqrt{k}$. In this Appendix we prove this statement.

First, we observe that in this solution, since both the elements of the filters and those of the inputs are non-negative, the networks behaves effectively as a linear operator. In particular, each layer corresponds to the application of a $L \times L$ circulant matrix $M$, which is obtained by stacking all the $L$ shifts of the following row vector,

$$[\underbrace{1, 1, ..., 1}_{F} \underbrace{0, 0, 0, ..., 0}_{L-F}]. \tag{10}$$

with periodic boundary conditions. The first row of such a matrix is fixed as follows. If $F$ is odd the patch of size $F$ is centered on the first entry of the first row, while if $F$ is even we choose to have $(F/2)$ ones at left of the first entry and $(F/2) - 1$ at its right. The output $f_k$ of the layer $k$ is then the following: $f_k(\delta_i) = M^k \delta_i$.

**Proposition A.1** *Let's consider the $L \times L$ matrix $M$ and a given $L$ vector $\delta_i$, as defined above. For odd $F \geq 3$, in the limit of large depth $\tilde{K} \gg 1$ and large width $\tilde{L} \gg F\sqrt{\tilde{K}}$, we have that*

$$(M^k)_{ab}\delta_i = \frac{1}{2\sqrt{\pi}\sqrt{D^{(1)}}\sqrt{k}}e^{-\frac{(a-i)^2}{4D^{(1)}k}}, \qquad D^{(1)} = \frac{1}{12F}(F-1)^3, \tag{11}$$

*while for even $F$:*

$$(M^k)_{ab}\delta_i = \frac{1}{2\sqrt{\pi}\sqrt{D^{(2)}}\sqrt{k}}e^{-\frac{(v^{(2)}k+a-i)^2}{4D_F^{(2)}k}}, \qquad D^{(2)} = \frac{1}{12F}\left(F^3 - 3F^2 + 6F - 4\right), \tag{12}$$

*with $v^{(2)} = (1-F)/(2F)$.*

**Proof:**  The matrix $M$ can be seen as the stochastic matrix of a Markov process, where at each step the random walker has uniform probability $1/F$ to move in a patch of width $F$ around itself. We write the following recursion relation for odd $F$,

$$p_{a,i}^{(k+1)} = \frac{1}{F}\left(p_{a-(F-1)/2,i}^{(k)} + ... + p_{a,i}^{(k)} + ... + p_{a+(F-1)/2,i}^{(k)}\right), \tag{13}$$

and even $F$,

$$p_{a,i}^{(k+1)} = \frac{1}{F}\left(p_{a-F/2,i}^{(k)} + ... + p_{a,i}^{(k)} + ... + p_{a+(F/2-1),i}^{(k)}\right). \tag{14}$$

In any of these two cases, this is the so-called master equation of the random walk (Risken, 1996). In the limit of large image width $L$ and large depth $\tilde{K}$, we can write the related equation for the continuous process $p_i(a, k)$, which is called Fokker-Planck equation in physics and chemistry (Risken, 1996) or forward Kolmogorov equation in mathematics (Saloff-Coste & Bremaud, 2000),

$$\partial_k p_{a,i}^{(k)} = v\partial_a p_{a,i}^{(k)} + D\partial_a^2 p_{a,i}^{(k)}. \tag{15}$$

where the drift coefficient $v$ and the diffusion coefficient $D$ are defined in terms of the probability distribution $W_i(x)$ of having a jump $x$ starting from the location $i$

$$v = \int dx W_i(x)x, \qquad D = \int dx W_i(x)x^2. \tag{16}$$

In our case we have $W_i(x) = 1/F$ for $x \in [i - (F - 1)/2, i + (F - 1)/2]$ for odd $F$ and $x \in [i - F/2, i + F/2 - 1]$ for even $F$, yielding the solutions for the Fokker-Planck equations for even and odd $F$ reported in Eq. 11 and Eq. 12.

We can better characterize the limits of large image width $L$ and large network depth $\tilde{K}$ as follows. The proof relies on the fact that a random walk, after a large number of steps, converges to a diffusion process. Here the number of steps is given by the depth $\tilde{K}$ of the network. Consequently, we need $\tilde{K} \gg 1$. Moreover, we want that the diffusion process is not influenced by the boundaries of the image, of width $L$. The average path walked by the random walker after $\tilde{K}$ steps is given by $F\sqrt{K}$. Then, we require $F\sqrt{K} \ll L$.

□

## B    EXPERIMENTAL SETUP

All experiments are performed in PyTorch. The code with the instructions on how to reproduce experiments are found here: tinyurl.com/github-experiments.

### B.1    DEEP NETWORKS TRAINING

In this section, we describe the experimental setup for the training of the deep networks deployed in Sections 1, 2 and 3.

For CIFAR10, fully connected networks are trained with the ADAM optimizer and learning rate $= 0.1$ while for CNNs SGD, learning rate $= 0.1$ and momentum $= 0.9$. In the latter case, the learning rate follows a cosine annealing scheduling. In all cases, the networks are trained on the cross-entropy loss, with a batch size of 128 and for 250 epochs. Early stopping at the best validation error is performed for selecting the networks to study. During training, we employ standard data augmentation consisting of random translations and horizontal flips of the input images. On the scale-detection task, we perform SGD on the hinge loss and halve the learning rate to $0.05$. All results are averaged when training on 5 or more different networks initializations.

For ImageNet, we used pretrained models from Pytorch, `torchvision.models`.

### B.2    SIMPLE CNNS TRAINING

In this section we present the experimental setup for the training of simple CNNs introduced in Section 4, whose sensitivities to diffeomorphisms and Gaussian noise are shown in Fig. 7.

To learn task 1 we use CNNs with stride $s = 1$ and filter size $F = 3$. The width of the CNN is fixed to 1000 channels, while the depth to 12 layers. We use the Scale-Detection task in the version of Fig. 5 (b), with $\xi = 11$ and gap $g = 4$ and image size $L = 32$. For the training, we use $P = 48$ training points and Stochastic Gradient Descent (SGD) with learning rate 0.01 and batch size 8. We use weight decay for the $L_2$ norm of the filters weights with ridge 0.01. We stop the training after 500 times the interpolation time, which is the time required by the network to reach zero interpolation error of the training set. The goal of this procedure is to reach the solution with minimal norm. The generalization error of the trained CNNs is exactly zero: they learn spatial pooling perfectly. We show the sensitivities of the trained CNNs, averaged over 4 seeds, in the top panels of Fig. 7, where we also successfully test the predictions (Eq. 5, Eq. 7). We remark that to compute $G_k$ we inserted Gaussian noise with already the ReLU applied on, since we observe that without it we would see a pre-asymptotic behaviour for $G_k$ with respect to $A_k$.

Task 2 is learned using CNNs with stride equal to filter size $s = F = 2$. For the dataset, we use the block-wise version of the Scale-Detection task shown in Fig. 5 (c), fixing $\xi = 2^5$ and $L = 2^7$. We use 7 layers and 1000 channels for the CNNs. The training is performed using SGD and weight decay with the same parameters as in task 1, with $P = 2^{10}$ training points. In the bottom panels of Fig. 7 we show that the predictions (Eq. 8, Eq. 9) capture the experimental results, averaged over 10 seeds.

To support the assumption done in Section 4 that the trained CNNs are effectively behaving as one channel with homogeneous positive filters, we report the numerical values of the average filter over channels per layer in Table 1 for Task 1 and Table 2 for Task 2. They are positive in the first 9 hidden layers, where channel pooling is most pronounced.

| | Init. | After training |
|---|---|---|
| $k = 1$ | $[0.0132, 0.0023, -0.0068]$ | $[0.2928, 0.2605, 0.2928]$ |
| $k = 2$ | $[0.0014, -0.0007, -0.0009]$ | $[0.0039, 0.0035, 0.0039]$ |
| $k = 3$ | $[-0.0006, -0.0001, 0.0010]$ | $[0.0043, 0.0038, 0.0043]$ |
| $k = 4$ | $[3.4610e - 05, 6.5687e - 04, -9.1634e - 04]$ | $[0.0039, 0.0033, 0.0038]$ |
| $k = 5$ | $[-0.0006, 0.0002, -0.0009]$ | $[0.0038, 0.0032, 0.0038]$ |
| $k = 6$ | $[0.0012, -0.0011, -0.0003]$ | $[0.0038, 0.0031, 0.0038]$ |
| $k = 7$ | $[-0.0006, 0.0004, 0.0003]$ | $[0.0041, 0.0032, 0.0040]$ |
| $k = 8$ | $[0.0005, -0.0012, 0.0010]$ | $[0.0036, 0.0024, 0.0035]$ |
| $k = 9$ | $[0.0005, -0.0012, 0.0010]$ | $[0.0021, 0.0016, 0.0017]$ |
| $k = 10$ | $[-0.0025, 0.0015, -0.0006]$ | $[-0.0013, -0.0008, -0.0010]$ |
| $k = 11$ | $[-0.0006, 0.0005, 0.0009]$ | $0.0002, 0.0002, 0.0002]$ |
| $k = 12$ | $[3.3418e - 04, 3.3521e - 05, 1.3936e - 03]$ | $[0.0009, 0.0008, 0.0009]$ |

Table 1: Average over channels of filters in layer $k$, before and after training, for simple CNNs with $s = 1$ and $F = 3$ trained on task 1. The network learns filters which are much more homogeneous than initialization.

| | Init. | After training |
|---|---|---|
| $k = 1$ | $[-0.0559, -0.0291]$ | $[0.3828, 0.3737]$ |
| $k = 2$ | $[-0.0022, 0.0010]$ | $[0.0060, 0.0059]$ |
| $k = 3$ | $[0.0006, -0.0010]$ | $[0.0064, 0.0065]$ |
| $k = 4$ | $[-0.0020, 0.0009]$ | $[0.0059, 0.0060]$ |
| $k = 5$ | $[0.0002, 0.0008]$ | $[9.9935e - 05, 2.1380e - 04]$ |
| $k = 6$ | $[-0.0003, -0.0010]$ | $[-0.0028, -0.0029]$ |
| $k = 7$ | $[-7.4610e - 04, 8.4595e - 05]$ | $[-0.0009, -0.0009]$ |

Table 2: Average over channels of filters in layer $k$, before and after training, for simple CNNs with $s = F = 2$ trained on task 2. The network learns filters which are much more homogeneous than initialization.

## C    ADDITIONAL FIGURES AND TABLES

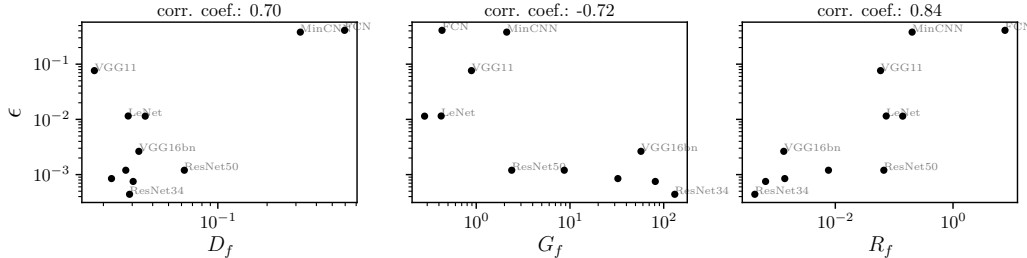

Figure 8: Generalization error $\epsilon$ versus sensitivity to diffeomorphisms $D_f$ (left), noise $G_f$ (center) and relative sensitivity $R_f$ (right) for a wide range of architectures trained on scale-detection task 1 (train set size: 1024, image size: 32, $\xi = 14, g = 2$). As in real data, $\epsilon$ is positively correlated with $D_f$ and negatively correlated with $G_f$. The correlation is the strongest for the relative measure $R_f$.

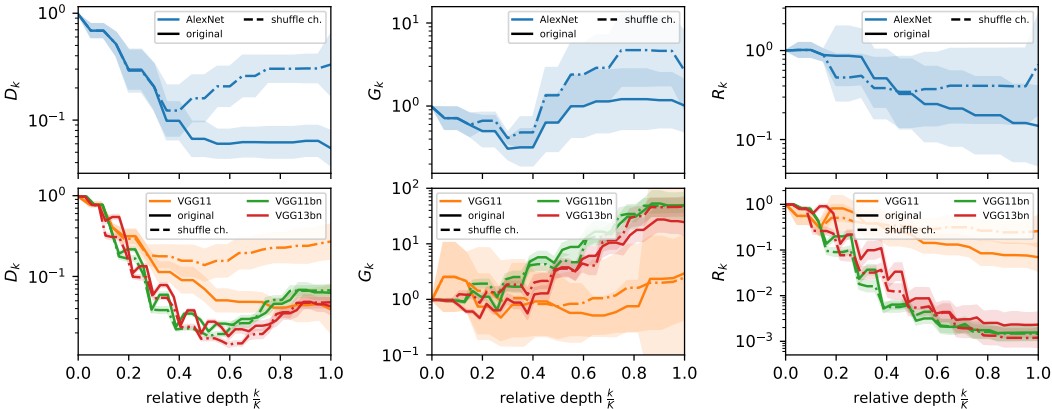

Figure 9: Sensitivities ($D_k$ left, $G_k$ middle and $R_k$ right) of the internal representations vs relative depth for AlexNet (1st row) and VGG networks (2nd row) trained on scale-detection task 1. Dot-dashed lines show the sensitivities of networks with shuffled channels.

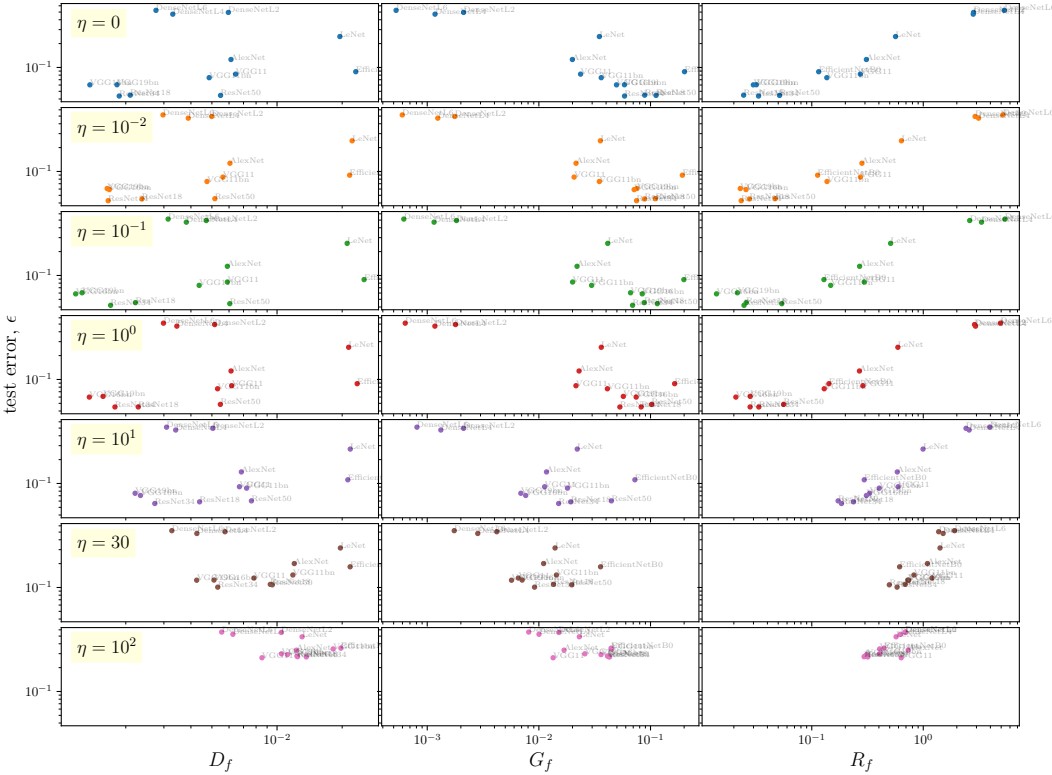

Figure 10: Test error vs. sensitivities (columns) when training on noisy CIFAR10. The different rows correspond to increasing noise magnitude $\eta$. Different points correspond to networks architectures, see gray labels. The content of this figure is also represented in compact form in Fig. 1, right.

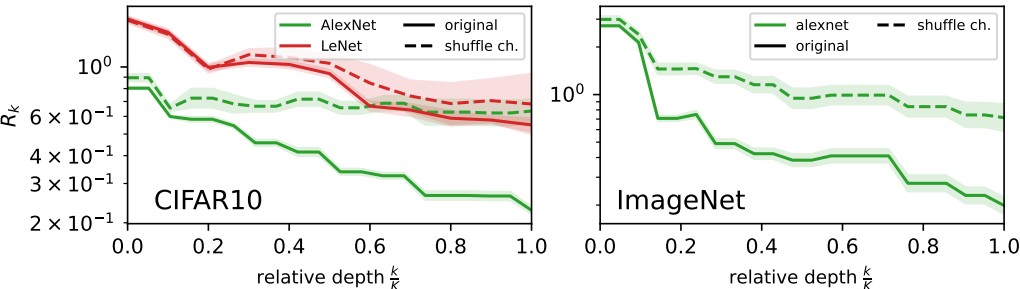

Figure 11: Analogous of Fig. 3 for different network architectures: relative sensitivity $R_k$ as a function of depth for LeNet and AlexNet architectures trained on CIFAR10 (left) and ImageNet (right). Full lines indicate experiments done on the original networks, dashed lines the ones after shuffling channels. $K$ indicates the networks total depth.

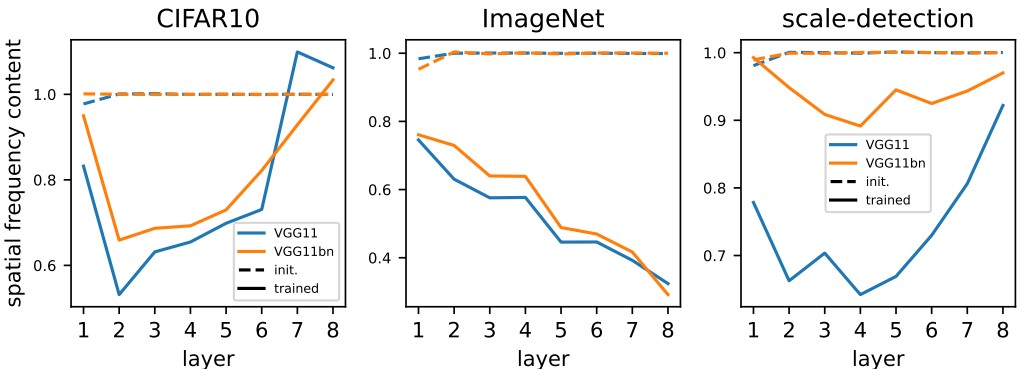

Figure 12: Spatial frequency content of filters for CIFAR10 (left), ImageNet (center) and the scale-detection task (right). The $y$-axis reports an aggregate measure among spatial frequencies: $N(\sum_{i=1}^{N} \lambda_l)^{-1} \langle \|\boldsymbol{w}_c^k\|^2 \rangle_c^{-1} \sum_{l=1}^{F^2} \lambda_l \langle (\boldsymbol{\Psi}_l \cdot \boldsymbol{w}_c^k)^2 \rangle_c$, where $\boldsymbol{\Psi}_l$ are the $3 \times 3$ Laplacian eigenvectors and $\lambda_l$ the corresponding eigenvalues, $\boldsymbol{w}_c^k$ the $c$-th filter of layer $k$ and $\langle \cdot \rangle_c$ denotes the average over $c$. This is an aggregate measure over frequencies, the frequencies distribution is reported in the main text, Fig. 4.

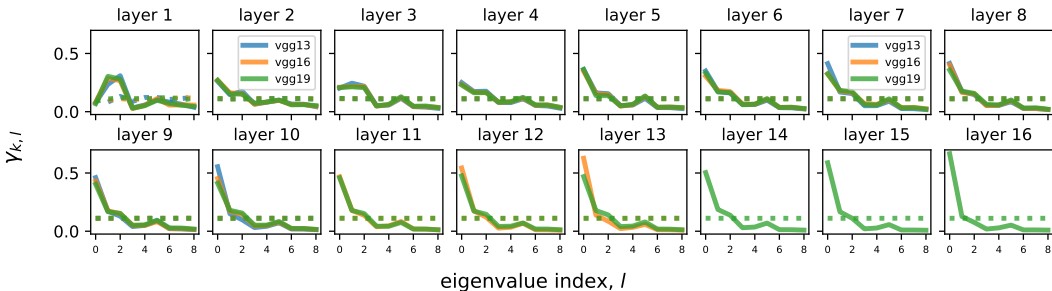

Figure 13: Analogous of Fig. 4 for deep VGGs trained on ImageNet. Dotted and full lines respectively correspond to initialization and trained networks. The $x$-axis reports low to high frequencies from left to right. Deeper layers are reported in rightmost panels.

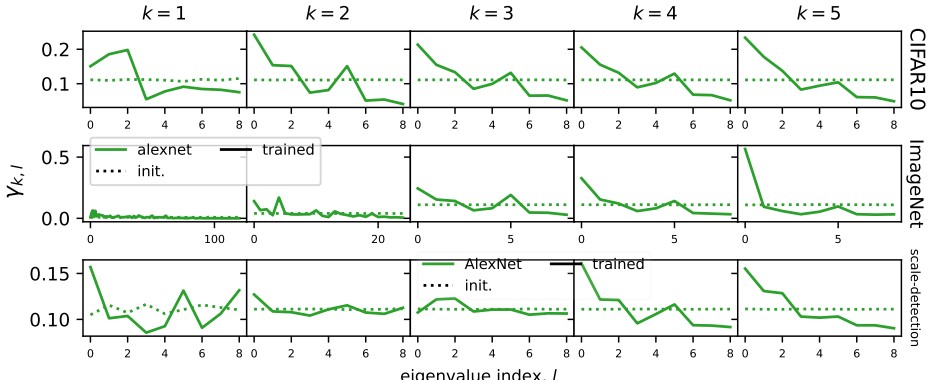

Figure 14: Analogous of Fig. 4 for AlexNet trained on CIFAR10 (1st row), ImageNet (2nd row) and the scale detection task (3rd row). Dotted and full lines respectively correspond to initialization and trained networks. The $x$-axis reports low to high frequencies from left to right. Deeper layers are reported in rightmost panels.

