# OpenReview forum: "How deep convolutional neural networks lose spatial information with training"
_ICLR.cc/2023/Conference — Submitted to ICLR 2023_

### Official Review · Reviewer_x5ja · 2022-10-21

**Confidence:** 3
**Correctness:** 2
**Technical Novelty And Significance:** 2
**Empirical Novelty And Significance:** 2
**Recommendation:** 3

**Clarity, Quality, Novelty And Reproducibility:**

- The paper includes code for reproducing the results.
- The paper is relatively easy to follow, though I do have some questions regarding the synthetic tasks and theoretical analysis:
    1. The proof for Section 4.1 seems to rely on that the number of layers k and the image size L both being sufficiently large. Is there a characterization how large this needs to be have this statement reasonably accurate? In particular, does it match the same regime of the empirical studies on CIFAR-10 (k < 20, L=32)?
    2. It is mentioned a few times in the arguments that the learned filters are non-negative, why is it the case?
    3. Does the theoretical analysis apply equally well to a trained neural network and an untrained (randomly initialized) network?

**Strength And Weaknesses:**

**Strength**
1. This paper identifies a clean question to study
2. This paper includes code to reproduce the experiments

**Weakness**
1. The empirical studies are weak
    1. The neural networks tested in this paper are very simple. Including some more modern networks would make the study more useful. At the minimum a deep resnet should be included, because skip connections are omnipresent in modern neural networks and it would be interesting to see how the arguments about the decoupling between filter and channel pooling behave with skip connections.
    2. What happen to other networks if you do the same kind of analysis as in Figure 4? I could not find such results in the Appendix either.
    3. It would be useful to add error bars to the plots (e.g. Fig. 3).
    4. Please include more than one dataset so that we get an idea of how generalizable the observations are.
    5. For many convnet architectures, the number of channels tend to increase at higher layers. In the experiments showing that lower layers are less sensitive to channel permutation, how does the number of channel impact the results?

2. The synthetic tasks are specifically designed to encourage network insensitivity to translations, and in task 2 the neural network architecture (strides equals to filter size) is also tailored to match the task bias. I'm having a hard time to see how this analysis here could help understand what happens in the real world scenarios.

**Summary Of The Paper:**

This paper studies the question of why trained deep convolutional networks are insensitive to image perturbations such as small translation and rotation. Through a set of studies on CIFAR-10, the paper argues that this is mostly due to spatial pooling and channel pooling, and the two components can be further decoupled at different layers. The paper then propose a simple synthetic task on which neural networks demonstrate similar behavior, and use this synthetic task to theoretically study the arguments made earlier.

**Summary Of The Review:**

This paper studies the question of why deep convolutional networks are insensitive to image perturbations such as translation. I find the empirical studies weak and the theoretical studies need some explanations (see above) to both clarify the results and to make connection to the empirical studies.

-------
After rebuttal: thanks to the authors for additional experiments on ImageNet. This addition to the paper require another round of careful review. And my other concerns (e.g. regarding the parameter regime of the proof and the justification for the non-negativity) are not well addressed by the response. Therefore I'm keeping my original rating.

---

> ### Author Response · Authors · 2022-11-14
> **Reply to Reviewer x5ja**
>
> We thank the Reviewer for their comments and suggestions.
>
> - On the empirical studies:
>     - **Comment.** The study of the role of skip connections in losing spatial information would be an interesting extension of this work but it goes beyond its present scope. Residual connections are equivalent to performing convolutions with very high-frequency filters. As a consequence, performing spatial pooling by a block (convs + skip conn.) does not necessarily involve just making convolutional filters low frequency but high-frequency filters may need to be built in order to balance the effect of skip connections. Despite that, we checked R_k for ResNet18 trained on ImageNet and the results are qualitatively consistent with the other results. We do not plan to include this result because of the complexity just mentioned. Yet, if the Reviewer insists, we will include it.
>     - **Action.** We now report results regarding the frequency content of filters from more networks both in Fig. 4 and in the new Fig. 13, 14.
>     - **Action.** Following the Reviewer’s suggestions, we added error bars to the measurements of R_k. Error bars report the standard deviation after averaging over a batch of test data points and different initializations of the architectures. The latter is only in the case we train the networks (i.e. for CIFAR10 and scale-detection).
>     - **Action.** We replicated our experiments on ImageNet. See also point [°] in the reply to Reviewer Zxqq.
>     - **Comment.**  Channel shuffling destroys the contribution of channel pooling to reducing R_k, independently of the number of channels.
>     - **Action.** We now better clarified the definition of channel pooling we provide in the text, see also discussion with Reviewer Zxqq.
>     - **Comment.**  The role of the synthetic task is to isolate the phenomenon that we want to understand, channel pooling, in order to quantify its role in building up the insensitivity to diffeomorphisms and sensitivity to Gaussian noise. Please refer also to the answer to Reviewer Zxqq, in particular to the point [°°].
> - **Comment.**  The proof relies on the fact that a random walk, after a large number of steps, converges to a diffusion process. Here the number of steps is given by the depth $\tilde{K}$ of the network. Consequently, we need $\tilde{K}\gg1$.
> Moreover, we want that the diffusion process is not influenced by the boundaries of the image, of width $L$. The average path walked by the random walker after $\tilde{K}$ steps is given by $F\sqrt{K}$. Then, we require $F\sqrt{K}\ll L$.
> Since the typical filter size and depth of the networks used on CIFAR-10 is $F=3$ and $K\sim 10$, and the image size is $L=32$, we are reasonably within these limits.
> - **Action.** We have included these characterizations in the new version of the manuscript, and we would like to thank the reviewer for pointing these out.
> - **Comment.**  The filters are taken non-negative since we find so after a direct inspection of the solutions reached by the CNNs after training. Please refer to the reply to Reviewer Zxqq for further details, in particular to the point [°°°].
> - **Comment.**  We thank the reviewer for the interesting question. We point out that the subject of convolutional networks at random initialization has already been studied in the literature. In particular, [1,2] find that the internal representation $f_k(x)$ of convolutional network behaves, in the limit of a large number of channels and width (and small bias), as Gaussian random fields with correlation matrix $\mathbb{E}\left[  f_k(x) f_k(y) \right] \approx \delta(x-y)$, with $x,y$ being input data and $\delta$ the Dirac delta.
> This spiky correlation matrix implies that for any perturbation $y=x+\varepsilon$, the representation $f_k(y)$ changes with respect to $f_k(x)$ independently on $\varepsilon$. This behavior is remarkably different to the smooth case achieved by the diffusion, after training. Consequently, both $D_k$ and $G_k$ are constant with respect to $k$ at initialization. This is consistent with the observations reported in the now Fig. 7.
> - **Action.** We added this comparison with the representation at initialization in the revised manuscript.
>
> [1] Lechao Xiao, Yasaman Bahri, Jascha Sohl-Dickstein, Samuel S Schoenholz, and Jeffrey Pennington. Dynamical isometry and a mean field theory of cnns: How to train 10,000-layer vanilla convolutional neural networks. ICML, 2018.
>
> [2] Schoenholz, S. S., Gilmer, J., Ganguli, S., and SohlDickstein, J. Deep Information Propagation. ICLR, 2017.

---

### Official Review · Reviewer_eGsy · 2022-10-22

**Confidence:** 4
**Correctness:** 3
**Technical Novelty And Significance:** 3
**Empirical Novelty And Significance:** 3
**Recommendation:** 6

**Clarity, Quality, Novelty And Reproducibility:**

The manuscript is clear and well written. Maybe  devoting the first figure to the illustration of the results of another paper (Petrini 2021) is not fully appropriate and might be partially misleading. In my opinion the most relevant results are those of section 4, which is very squeezed (for example fig 9 is almost not discussed).

**Strength And Weaknesses:**

Strengths:
1.	Investigating the working principle of deep CNNs remains a topic of interest, and the manuscript sheds some light on the different roles of special and channel pooling
2.	The results on the artificial task, section 3 and 4, are neat. In particular, the analytical derivation in section 4 is enlightening with respect to the manner in which noise is processed in a CNN
Weaknesses:
1.	The empirical results presented in support to the first “contribution”, disentangling spacial and channel pooling, are not fully convincing. Fig 3 shows the changes in the sensitivity to shuffling the channel connections. The scale is linear in the left panel (AlexNet and LeNet) and log in the right panel (VGG) creating some confusion.  The change is claimed to be small in the first layers, but this change looks approximately 30 % also for very deep layers in VGG. Only in LeNet the change is clearly small (~ 5 %). Is 30 % small or large? Maybe a baseline should be defined. Moreover, the qualitative differences between the different architectures are not explained. Do they derive from the depth?
2.	Fig. 4 illustrates the frequency content of the filters of the layers of a CNN at different depth. I do not understand why the curves do not all start from 1. Moreover, I do not understand what we learn from the time dependence of these coefficient.  The main message conveyed by this figure should be that deep layers (1-5) “become low-frequency” (while, I assume, late layers become high frequency). A better observable would possibly be the average frequency of the filter\ sum_{kl} gamma_kl omega. I also have a problem with the interpretation of the frequency content as defined by eq. 3. The receptive field becomes larger and large with depth. Therefore, the same frequency in different layers corresponds to different physical lengths in the input image. Shouldn’t the frequency be mapped in common units across the layers? I also do not fully understand why the observation that early kernels are "low-frequency" supports the claim that channel pooling happens late in the network.


**Summary Of The Paper:**

The manuscript analyzes how convolutional neural networks (CNNs) process spatial information present in the images. The analysis builds on the observation in Petrini 2021 that performance of CNNs is correlated with their invariance towards diffeomorphisms, namely “smooth” transformations of the images,  and anti-correlated with their sensitivity to noise added to the images. The specific goal of this paper is disentangling the role of spatial pooling and channel pooling, operations which are at the basis of the working principle of CNNs. The analysis is performed using real data (CIFAR10) in Sec 2 and synthetic tasks in Sec 3-4.

**Summary Of The Review:**

The manuscript provides some interesting and novel insight on the manner in which CNNs process the information. The analysis of a very simple synthetic dataset presented in section 3 and 4 is clear, while the analysis of CIFAR10 (section 2) is in my opinion not fully convincing. I think the contribution is valuable, but some points need clarification.

---

> ### Author Response · Authors · 2022-11-14
> **Reply to Reviewer eGsy**
>
> We thank the Reviewer for their comments and suggestions. For the discussion regarding Fig. 3, please see the answer to Reviewer Zxqq.
> - **Action.** As suggested, we modified Fig. 4 and removed the time dependence, which indeed makes the figure easier to parse. We highlight that layer 2-6 become low frequency, while a clear bias toward high/low frequencies does not emerge for layers 1, 7, 8.
> - **Comment.** In the previous version of Fig. 4, curves did not start from 1 as the first point shows the frequency content of filters after one epoch of training.
> - **Action.** Following the Reviewer suggestion, we added one figure showing an aggregate measure of the frequency content of filters per layer (see Figure 11 in Appendix). This is a weighted average frequency, normalized such that it is equal to 1 if all frequencies are equally represented.
> - **Comment.** Regarding the discussion on frequencies at deeper layers corresponding to larger and larger length-scales in input space, we remark that we are interested in investigating the relative relations between frequencies within a given layer, more than their absolute value. In simple terms, we want to highlight that filters are nearly homogeneous, irrespective of the depth.
> - **Comment.** We agree with the reviewer that early kernels being low-frequency does not imply that channel pooling only happens late in the network and we have removed such a statement from the manuscript (see also reply to reviewer Zxqq).
> - **Comment.** We remark that only the central plot in Fig. 1 is adapted from another paper while the right panel shows new data. We decided to include it, while properly citing the reference, as it serves as the main motivation for our work.
> - **Action.** We expanded the discussion on Fig. 9 (Fig. 7 in the revised pdf).

---

### Official Review · Reviewer_Zxqq · 2022-10-24

**Confidence:** 4
**Correctness:** 2
**Technical Novelty And Significance:** 2
**Empirical Novelty And Significance:** 2
**Recommendation:** 3

**Clarity, Quality, Novelty And Reproducibility:**

Clarity: The paper is mostly well-written, however some parts need more clarification (see especially my last comment on Weaknesses).
Quality: In my opinion, the experiments are insufficient in supporting the claims of the paper convincingly. Authors use very strong wording in many cases that is not completely justified by the experiments.
Novelty: the direction of the empirical investigation is somewhat novel to the best of my knowledge, however the main phenomenon related to invariance to diffeomorphisms and sensitivity to noise has been investigated in prior work.
Reproducibility: code to reproduce the results has been provided.
Minor comments:

- typo in Introduction "Here the inputs images"

- typo in Section 1.2 "in the context adversarial robustness"

**Strength And Weaknesses:**

Strengths:

- The paper is organized and easy to follow. The structure is logical and the paper is mostly well-written.

- The problem the paper investigates is very interesting and is important in better understanding the success of modern deep learning. Overall, the motivation of the paper is clear.


Weaknesses:

- In my opinion, the main claims of the paper are not well supported by the experiments. Based on a single small dataset (CIFAR-10) authors conclude that "spatial and channel pooling are carried out along the whole network", however it is not clear to me how authors arrive at this conclusion. The decrease in sensitivity to diffeomorphisms and increase to white Gaussian noise may be caused by some other factors not investigated in this paper. The proposed underlying reasons (different types of pooling at different depths) are a possible and logical explanation, but I don't see how this simple experiment proves this. I am also not convinced by the experiments disentangling the effect of the two types of pooling.

  - First, in Figure 3 right, in case of VGG11bn the gap between original and channel-shuffled models is already significant in early layers, and in case of VGG11 the gap is almost constant in the first 80% of the network. Overall, the presented evidence is not sufficient to conclude (especially based on a single small dataset) that channel pooling is not happening early in the network.

  - Second, the plots showing the frequency distribution of learned filters with respect to spatial pooling are difficult to make out. Authors claim that "layers 2 to 5 become low-frequency with training", however the lines are so close to each other that it is impossible to verify this. Probably presenting the distribution at t=0 and at end of training would be much easier to interpret, the exact values during training seem to be irrelevant to the paper. Moreover, even if the claim is true, why do we not see pooling at layer 1?

- It is not clear to me what is the goal of defining the scale-detection tasks. Authors concluded based on standard image classification that the phenomenon holds (early spatial pooling, late channel pooling). Then, they design a problem where our intuition suggests that pooling is the simplest solution, and we observe the phenomena again. I don't see what the scale-detection tasks add that we did not see in the image classification experiments.

- With respect to task 1, authors claim that "Spatial pooling is the most direct mean to achieve such insensitivity" with respect to sensitivity to diffeomorphism. This might be true based on our human intuition, but the learning dynamics of neural networks is not very well understood and this claim sounds too strong.

- The results on task 1 are not very convincing in Figure 7. In particular, it looks like VGG11's sensitivity to Gaussian noise is more or less constant over layers. The phenomenon seems to hold for networks with batch normalization only.

- The theoretical analysis is a bit confusing to me. What diffeomorphisms have been used? Why is the translation only 1 pixel? Why are filters necessarily non-negative?

**Summary Of The Paper:**

This paper investigates the mechanism of spatial information loss in image classification networks. This loss in sensitivity to small perturbations in the input is important in learning robust invariant representations of features relevant to the task (image classification in this work). Authors propose that deep convolutional networks learn to first spatially and then channel-wise pool information as we go deeper in the network layer-by-layer. They support their claims with numerical experiments on the CIFAR-10 dataset on multiple architectures. Then, they construct some simple vision tasks, where the simplest solution according to human intuition is pooling, and they demonstrate that the above phenomena from image classification still hold. Finally, they quantify in this simple toy case how sensitivity to input diffeomorphism and additive white noise scales with network depth.

**Summary Of The Review:**

Overall, I would lean towards rejecting the paper in its current state. I think the empirical evidence provided is not strong and convincing enough due to the reasons I have detailed above under 'Weaknesses'. Furthermore, the motivation for introducing the scale-detection tasks and the theoretical analysis is not clear to me.

---

> ### Author Response · Authors · 2022-11-14
> **Reply to Reviewer Zxqq (Part 1/2)**
>
> We would like to thank the reviewer for the insights. Addressing the weakness remarked by the reviewer:
> - **Comment.** First of all, we would like to clarify what we mean by spatial and channel pooling. Spatial pooling is the integration of local patches and can be realized in a single convolutional layer by having homogeneous filters. Channel pooling refers broadly to any operation that decreases the sensitivity of the representation by combining the outputs of different channels. In this respect, it is obviously true that as long as the sensitivity of the representation keeps decreasing, spatial and/or channel pooling must occur.
> - **Action.** We have revised the manuscript to make the definition of channel pooling clear. See in particular the third paragraph of the introduction and the first three paragraphs of Section 2.
> - **Comment.** Using this definition, it is clear that shuffling the channels destroys any possible contribution of channel pooling. The fact that the sensitivity to diffeomorphisms still decreases after shuffling the channels proves that spatial pooling is being performed.
> - **Action.** We agree with the reviewer that this does not imply that channel pooling is not being performed in the first half of the network, therefore we have removed statements of this kind from the manuscript (see e.g. first point of the `our contributions’ list). Moreover, we changed Fig. 4 according to the Reviewer's suggestion and now it is easier to parse. This change makes it clearer that layer 1 it not becoming low-frequency with training. This fact is consistent with the Reviewer’s observation that a gap between original and shuffled curves already opens at the first layer. We now state this point in the main text.
> - **Comment.** We agree with the necessity to check the robustness of our results on a larger dataset.
> - **[°] Action.** We replicated the experiments in Fig. 3 and 4 on ImageNet using pretrained models from PyTorch. We find that both spatial and channel pooling contribute to reducing $R_k$ for all $k$, except $k=1$. Consistently, all filters, except the ones of the first layer, become low frequency with training. We now include experiments on ImageNet in the main text, which we think is a significant addition to our paper.

---

> > ### Author Response · Authors · 2022-11-14
> > **Reply to Reviewer Zxqq (Part 2/2)**
> >
> > - **[°°] Comment.** The goal of defining the scale-detection task is to introduce a synthetic task which *(i)* captures the spatial pooling observed with real data and *(ii)* can be compared with our analytical treatment of the problem. The idea is to have a task which isolates the phenomena that we want to understand and it is simpler to analyze. This kind of modeling is omnipresent in physics and it allows identifying the basic principles behind natural phenomena. This approach allows us to clearly identify and quantify how insensitivity to diffeomorphisms and sensitivity to noise build up in the network thanks to spatial pooling.
> > - **Comment.** We agree that there is no guarantee that CNNs would reach the intuitive solution of spatial pooling to learn the scale-detection task. However, we argue that this is the case based on the following empirical observations:
> >     - In (now) Fig. 9 the sensitivities obtained before and after shuffling the channels of the VGGs trained on the scale-detection task are essentially the same. This suggests that interactions between channels play no role towards building insensitivity to diffeomorphisms.
> >     - In Fig. 4, bottom panels, we show that the constant mode is the dominant one within the learned convolutional filters for layers 2-7, which are the layers where $R_k$ decreases the most in Fig. 9. This suggests that spatial pooling is present.
> > - **Action.** We clarified this point in the new version of our work and modified the sentence the Reviewer finds problematic.
> > - **Comment.** On $G_k$ for VGG11 and the scale-detection task: as suggested by another reviewer we added error bars to the plots and noticed this specific measurement to be very noisy.
> > - **Action.** We trained more newly initialized networks to add statistics to the measurement. However, the large noise persists. We kept the curve in the figure for completeness (now Fig. 9).
> > - **Comment.** The diffeomorphisms used in Section 3 and Section 4 for the scale-detection task have indeed not been specified and we thank the reviewer for pointing this out. In Section 3, for two-dimensional images, each active pixel moves with probability 1/9 to either one of its 8 neighboring pixels or it stays where it is.  In Section 4, for one-dimensional images, each active pixel moves with probability ½ either at left or right. We decided to fix the length of these displacements to just 1 pixel because *(i)* is the smallest value that prevents the use of interpolation, which would mess up with the content of the inputs, i.e. one pixels becomes a blob and deformed images are not natural anymore *(ii)* makes the theoretical analysis of Section 4 easier. We added these specifications in the revised manuscript.
> > - **[°°°] Comment.** We assume the filters are non-negative because we find empirically that, in the layers where $R_k$ decreases, they are actually positive.
> > - **Action.** This is now shown in Table 1 and 2 of the revised manuscript.

---

### Author Response · Authors · 2022-11-14
**Manuscript revision updated**

We thank all reviewers for their insightful comments and constructive critiques.  We revised our manuscript accordingly, modifications are highlighted in red.

---

### Decision · Program_Chairs · 2023-01-20

**Decision:**

Reject

**Justification For Why Not Higher Score:**

There's really no support from any of the reviewers for this paper.  The study is limited in scope and the results are not clearly in support of the hypotheses

**Justification For Why Not Lower Score:**

n/a

**Metareview: Summary, Strengths And Weaknesses:**

This paper explores preexisting models and the stability of the models to small changes in input.  This paper aims to better characterize the kinds of stability offered by different networks at different levels.  Several experiments are performed to create an interesting picture of how  networks compute their representations.

The reviewers pointed out several strengths: the paper is well-organized and the motivation is strong. Reviewers agree this is an interesting problem to study. The reviewers pointed out weaknesses, including the lack of diversity amongst models and datasets.  How can these findings be generalized to other models and datasets?  A more thorough set of experiments is needed. The reviewers also wonder about the soundness of the findings, and if the experiments support the claims.

The authors did make some significant changes to the paper over the review period, and we thank them for their commitment to the reviewing process.